# Multiphase Computed Tomography Scan Findings for Artificial Intelligence Training in the Differentiation of Hepatocellular Carcinoma and Intrahepatic Cholangiocarcinoma Based on Interobserver Agreement of Expert Abdominal Radiologists

**DOI:** 10.3390/diagnostics15070821

**Published:** 2025-03-24

**Authors:** Nakarin Inmutto, Suwalee Pojchamarnwiputh, Wittanee Na Chiangmai

**Affiliations:** Department of Radiology, Faculty of Medicine, Chiang Mai University, Chiang Mai 50200, Thailand; nakarin.i@cmu.ac.th (N.I.); suwalee.poj@cmu.ac.th (S.P.)

**Keywords:** hepatocellular carcinoma, intrahepatic cholangiocarcinoma, artificial intelligence, computed tomography, interobserver agreement

## Abstract

**Background/Objective**: Hepatocellular carcinoma (HCC) and intrahepatic cholangiocarcinoma (ICC) are the most common primary liver cancer. Computed tomography (CT) is the imaging modality used to evaluate liver nodules and differentiate HCC from ICC. Artificial intelligence (AI), machine learning (ML), and deep learning (DL) have been used in multiple studies in the field of radiology. The purpose of this study was to determine potential CT features for the differentiation of hepatocellular carcinoma and intrahepatic cholangiocarcinoma. **Methods**: Patients with radiological and pathologically confirmed diagnosis of HCC and ICC between January 2013 and December 2015 were included in this retrospective study. Two board-certified diagnostic radiologists independently reviewed multiphase CT images on a picture archiving and communication system (PACS). Arterial hyperenhancement, portal vein thrombosis, lymph node enlargement, and cirrhosis appearance were evaluated. We then calculated sensitivity, specificity, the likelihood ratio for diagnosis of HCC and ICC. Inter-observed agreement of categorical data was evaluated using Cohen’s kappa statistic (k). **Results**: A total of 74 patients with a pathologically confirmed diagnosis, including 48 HCCs and 26 ICC, were included in this study. Most of HCC patients showed arterial hyperenhancement at 95.8%, and interobserver agreement was moderate (k = 0.47). Arterial enhancement in ICC was less frequent, ranging from 15.4% to 26.9%, and agreement between readers was substantial (k = 0.66). The two readers showed a moderate agreement of cirrhosis appearance in both the HCC and ICC groups, k = 0.43 and k = 0.48, respectively. Cirrhosis appeared in the HCC group more frequently than the ICC group. Lymph node enlargement was more commonly seen in ICC than HCC, and agreement between the readers was almost perfect (k = 0.84). Portal vein invasion in HCC was seen in 14.6% by both readers with a substantial agreement (k = 0.66). Portal vein invasion in ICC was seen in 11.5% to 19.2% of the patients. The diagnostic performance of the two radiologists was satisfactory, with a corrected diagnosis of 87.8% and 94.6%. The two radiologists had high sensitivity in diagnosing HCCs (95.8% to 97.9%) and specificity in diagnosing ICCs (95.8% to 97.9%). **Conclusions:** Cirrhosis and lymph node metastasis could be ancillary and adopted in future AI training algorithms.

## 1. Introduction

Hepatocellular carcinoma (HCC) and intrahepatic cholangiocarcinoma (ICC) are the most common primary liver cancers, accounting for approximately 85% [1]. These two cancers are a cause of a considerable part of oncologic mortality, and they are also a diagnostic and therapeutic challenge for healthcare systems worldwide. Underlying chronic hepatic inflammation with viral hepatitis B or C is a risk factor for HCC, but less for ICC. The correct diagnosis of HCC and ICC is crucial because the treatment of the two entities is different, and percutaneous biopsy is not widely available in Thailand. Rapid and reliable detection and diagnosis of HCC and ICC may allow for earlier treatment onset and better outcomes for these patients.

The role of imaging in the management of HCC and ICC has significantly evolved and expanded beyond the plain radiological confirmation of the tumor based on the typical appearance in a multiphase contrast-enhanced tomography (CT) examination. Both HCC and ICC have characteristic imaging appearances on multiphase CT and magnetic resonance imaging (MRI) [2]. CT is the imaging modality used to evaluate liver nodules and differentiate HCC from ICC. As cross-sectional imaging has become more widely available and of higher quality, the reliance on invasive diagnostic biopsies has declined, positioning imaging-based diagnosis as a more integral part of medical decision-making. This is because these two liver cancers have different enhancing patterns on multiphase CT. Typical HCC shows arterial hyperenhancement, washout appearance in the portovenous phase, and the presence of an enhancing capsule. Vascular invasion, particularly of the portal vein, is an imaging finding of advanced HCC. The American College of Radiology proposed the Liver Imaging-Reporting and Data System (LI-RADS) for standardized imaging diagnosis criteria in patients at risk of HCC [3]. This system is used as a system for patients developing HCC, and this algorithm guides radiologists in differentiating between benign and HCC masses. The LI-RADS is expected to become an international standard in the imaging diagnosis of liver masses. LI-RADS 5 lesions are diagnostic of HCC without the need for histologic confirmation, as shown in Figure 1. In contrast to HCC, typical ICC shows rim-like enhancement at the tumor periphery during the arterial phase and progressive centripetal enhancement in a later phase, as shown in Figure 2. Intrahepatic bile duct dilatation is another radiological finding of ICC. The characteristics and differences between typical HCC and ICC CT findings are described in Table 1.

However, an atypical enhancement pattern of HCC and ICC has often been seen in daily practice, making diagnosis of HCC or ICC challenging [4,5,6,7,8,9,10] (Figure 3). A liver nodule in a high-risk patient might have an enhancing pattern similar to ICC. On the other hand, small ICCs have enhancing patterns mimicking HCC, as shown in Figure 4. One study showed that the accuracy of CT for distinguishing non-HCC malignancies from HCC was 79.9% [11]. Therefore, the diagnosis and differential diagnosis of HCC and ICC is sometimes difficult and requires significant experience from radiologists.

Artificial intelligence (AI) was developed many years ago. Machine learning (ML) is a subset of AI whereby a system can develop the ability to “learn” from data without being explicitly programmed. There are three main types of ML: supervised, unsupervised, and reinforcement learning. Supervised learning uses labeled datasets to make classifications, predictions, and regression. Unsupervised learning uses unlabeled datasets to identify not readily apparent patterns and classify cases. Reinforcement learning is an area of machine learning that handles sequential decision-making problems in a situation of uncertainty. Reinforcement learning optimizes sequential decisions by finding the best strategy, divided into supervised and unsupervised methods [12].

Supervised learning deals with annotated data with input–output pairs and common techniques including linear regression, logistic regression, decision trees, k-nearest neighbor, support vector machine (SVM), random forest (RF), naive Bayes classification, and gradient boosting [13]. ML and deep learning (DL) have been used in multiple studies in the field of medicine. Multi-layered neural network algorithms, including convolutional neural networks (CNNs), have proven of great use in the analysis of radiology images [14,15]. Machine learning algorithms have demonstrated exceptional accuracy in classifying various diseases through radiological imaging. Two studies have investigated the usefulness of DL-based liver tumor classification from a multiphase contrast-enhanced CT scan [16,17].

AI models have been trained on CECT datasets to extract key imaging biomarkers that help distinguish HCC from ICC and other liver lesions. Several studies have demonstrated the effectiveness of AI in liver tumor classification. Yasaka et al. developed a CNN model that is capable of differentiating HCC, ICC, hemangiomas, and metastases based on multiphase CT, achieving a 92% area under the receiver operating characteristic curve (AUC) [18]. Similarly, Ponnoprat et al. applied deep learning techniques to multiphase CT scans, reporting an 88% classification accuracy in distinguishing HCC from ICC [18]. Wang et al. further advanced AI applications by proposing an interpretable radiomics-based system that identifies radiological features in liver tumors, demonstrating an 82.9% sensitivity in detecting HCC [19].

The noninvasive differentiation of HCC from ICC remains challenging [2,4,5,6,20]. Therefore, we attempted to determine the frequency and reliability of CT features that are commonly used by radiologists for the diagnosis of HCC and ICC. Then, we used these CT features to train the AI dataset for differential diagnosis HCC and ICC. Finally, we improved the accuracy of AI applications in the field of hepatology.

## 2. Materials and Methods

This retrospective study was conducted in accordance with the Declaration of Helsinki and was approved by the Research Ethics Committee, study code: RAD-2562-06135. Informed consent was not required due to the retrospective nature of this study.

### 2.1. Patient Selection

Patients with radiologically and pathologically confirmed diagnoses of HCC and ICC between January 2013 and December 2015 were included in this retrospective study. The diagnosis of ICC was confirmed based on histopathology (biopsy or resection). The diagnosis of HCC was confirmed based on radiological findings or histopathology. The patient selection flow chart is presented in Figure 5. The inclusion criteria were patients with a diagnosis of HCC and ICC in the hospital database, patients with a pre-diagnosis multiphase CT scan in a picture archiving and communication system (PACS), and pathologically confirmed diagnosis of HCC or ICC in the Pathology Department database. The exclusion criteria were poor images quality or improper phases of CT scan, the pathological diagnosis of metastasis or other malignancies, and CT or pathological report of perihilar or extrahepatic cholangiocarcinoma.

### 2.2. Imaging Acquisition

CT images of the liver were obtained using one of two multidetector CT scanners (SOMATOM FORCE and SOMATOM DEFINITION, Siemens Medical Solutions, Forchheim, Germany). The technique included pre-contrast imaging and multiphase post-contrast imaging following injection of 100 mL of non-ionic iodinated contrast media (350 mg/mL) through the antecubital vein. The hepatic arterial phase (HAP), portovenous phase (PVP), and delayed phase (DP) of scanning began 35, 80, and 180 s after the injection of the contrast media, respectively. The images were reconstructed in 2 mm slices in axial, coronal, and sagittal views.

### 2.3. Imaging Interpretation

CT images were independently reviewed using Fuji PACS and using Synapse workstation version 5, https://www.fujifilm.com/th/en/healthcare/healthcare-it/it-imaging/pacs, accessed on 6 March 2025, by two expert board-certified Diagnostic Radiologists, Suwalee Pojchamarnwiputh and Wittanee Na Chiangmai, who had more than 20 years of experience in abdominal radiology. The readers were blinded to both radiological and pathological reports. Each reader independently viewed and evaluated the images, then measured the tumor size on the most visible imaging phase. Each tumor was evaluated in terms of arterial hyperenhancement, portal vein invasion, and final diagnosis. The presence of cirrhosis and lymph node metastasis was evaluated.

The following formal radiological criteria were applied:

Arterial hyperenhancement: Defined as non-rim, unequivocal enhancement of the lesion compared to the surrounding liver parenchyma during the arterial phase, according to the Liver Imaging Reporting and Data System (LI-RADS) guidelines.

Portal vein invasion: Defined as the presence of a soft tissue thrombus or filling defect within the portal vein, with or without vein expansion, showing enhancement characteristics similar to the tumor on post-contrast imaging.

Cirrhosis: Identified based on morphological features such as surface nodularity, lobar or segmental volume redistribution (e.g., caudate lobe hypertrophy or right lobe atrophy) and secondary signs of portal hypertension (e.g., splenomegaly, portosystemic collaterals).

Lymph node metastasis: Defined as lymph nodes with a short-axis diameter of >1 cm or nodes exhibiting central necrosis, irregular margins, or abnormal enhancement, located in periportal, peripancreatic, or para-aortic regions.

### 2.4. Histopathologic Examination

The histopathologic report served as the reference standard of this study. Biopsy or surgical specimens of the hepatic nodule were fixed with 10% formalin and embedded in paraffin. The tissue slices were stained with hematoxylin–eosin and evaluated by pathologists. No reexamination of pathological slices was done.

### 2.5. Statistics

Statistical analyses were performed using the software SPSS (IBM SPSS Statistics for Windows, Version 19. Armonk, NY, USA: IBM Corp).

Quantitative data were presented as means and standard deviations or medians and interquartile ranges depending on the data distribution. Categorical variables were analyzed using the chi-square test. The arterial hyperenhancement, portal vein thrombosis, lymph node enlargement, and cirrhosis appearance were calculated and presented with the sensitivity, specificity, and likelihood ratio for diagnosis of HCC and ICC.

Inter-observed agreement of categorical data was evaluated using Cohen’s kappa statistic with corresponding 95% confidence intervals. According to Landis and Koch, values of less than 0.00 indicate no agreement, 0.01–0.20 indicate slight agreement, 0.21–0.40 indicate fair agreement, 0.41–0.60 indicate moderate agreement, 0.61–0.80 indicate substantial agreement, and 0.81–1.00 indicate almost perfect agreement [21]. Significance was defined as *p* < 0.05.

## 3. Results

### 3.1. Participants

A total of 74 patients with a pathologically confirmed diagnosis, including 48 HCCs and 26 ICC, were included in this study. The mean age of patients with HCC was 52 years (range 21–76), while the mean age of those ICC was 57 years (range 35–76).

### 3.2. CT Features of HCC and ICC and Interobserver Agreement

Frequent CT features and interobserver agreement between the two readers are demonstrated in Table 2 and Table 3. Arterial enhancement in HCC was observed in 95.8% of cases by both readers. The agreement between the readers was moderate (k = 0.47). Arterial enhancement in ICC was identified less often, in 15.4% of cases by Reader 1 and 26.9% by Reader 2. The agreement between the readers was substantial (k = 0.66).

The two readers showed moderate agreement of cirrhosis appearance in both HCC and ICC. Cirrhosis appeared in the HCC group more frequently than in the ICC group (70.8–72.9% vs. 42.3–61.5%)

Lymph node enlargement in HCC was not commonly seen, in 14.6% of the cases by Reader 1 and 25.0% by Reader 2. The agreement between the readers was fair (k = 0.29). Lymph node enlargement in ICC was more commonly seen, in 50.0% of the cases by reader 1 and 57.7% by Reader 2. The agreement between the readers was almost perfect (k = 0.84).

Portal vein invasion in HCC was seen in 14.6% of the cases by both readers with substantial agreement (k = 0.66). Portal vein invasion in ICC was seen in 11.5% of the cases by Reader 1 and 19.2% by Reader 2. The agreement of both readers was substantial agreement (k = 0.70).

### 3.3. Correct Diagnosis

The diagnostic performance of the two radiologists was satisfactory, with a corrected diagnosis of 87.8% and 94.6% and a *p* value of 0.108. Both radiologists shared a common correct diagnosis of 65 out 74 patients, including 46 HCCs and 19 ICCs. Two HCCs and 7 ICCs, which were 4.15% of HCCs and 26.9% of ICCs, were misdiagnosis by both radiologists. The diagnostic performance of the two expert radiologists in the diagnosis of HCC and ICC is summarized in Table 4.

## 4. Discussion

According to LI-RADS, the percentage of HCC detected based on LR-5 criteria is 95%, while the percentage of malignancy is 98% [3]. Most HCC cases showed arterial hyperenhancement (95.8%), and interobserver agreement was moderate. The presence of arterial enhancement in our study was higher than in prior studies, likely due to the HCCs in our study being larger than in other studies. The presence of arterial enhancement was related to arterial blood flow, which increases in the early stage of hepatocarcinogenesis in well-differentiated and moderately differentiated HCC. Poorly differentiated HCCs have decreased arterial blood flow, resulting in a typical enhancement pattern of some HCCs.

On the other hand, some ICCs show arterial enhancement. Using the presence of arterial hyperenhancement alone may lead to misdiagnosis. Interobserver agreement regarding presence or absence of arterial enhancement was substantial.

Two expert radiologists had high sensitivity in diagnosed HCCs, with a high positive likelihood ratio and substantial agreement. In addition, ICCs were diagnosed with lower sensitivity but higher specificity and a positive likelihood ratio.

Cirrhosis can be associated with an over 30-fold increase in HCC risk and a 10-fold to 20-fold increase in ICC risk. Cirrhosis appearance has been used as a radiological finding for the prediction of HCC. Our study demonstrated cirrhotic appearance in patients with HCCs more than in patient with ICCs; around 70% versus 40 to 60%. Interobserver agreement of cirrhotic appearance was moderate. We favored diagnosing HCC in patients with cirrhotic appearance.

Lymph node enlargement or lymph node metastasis is more commonly seen in ICC than in HCC. Our study revealed lymph node metastasis in about 50% of ICC patients, with almost perfect agreement between the two expert radiologists. We almost diagnosed ICCs when CT showed lymph node enlargement.

Machine learning methods have been widely adopted in classifying liver cancer [13,15,17,18,19,22,23,24,25,26]. Supervised learning with a set of data labeled for algorithms such as support vector machines (SVMs) has been used in differentiated liver cancer. Ponnoprat et al. achieved 88% classification accuracy of differentiated HCCs and ICCs based on multiphase CT scans [16]. Deep learning is currently being investigated for liver cancer classification. There are still several limitations due to selection bias of training datasets, small sample sizes, and lack of internal and external validation. A convolutional neural network (CNN) is a kind of deep learning method that does not require the definition of specific radiological features to learn. It interprets a large of training images and may discover additional differential features not yet identified in current radiological practice. Yasaka used a CNN algorithm for differentiation of liver masses into five classes and found that the median area under the receiver operating characteristic curve for differentiating malignant and benign lesions was 0.92 [18].

Wang et al. proposed a proof-of-concept interpretable deep learning system for clinical radiology by identifying and scoring radiological features [27,28,29]. They used 14 radiological features such as arterial phase hyperenhancement, central scar, enhancing rim, progressive centripetal filling, progressive hyperenhancement, thin-walled mass, and washout pattern to classify liver lesions into six classes: benign cyst, cavernous hemangioma, focal nodular hyperplasia, HCC, ICC, and metastasis. Their model achieved 76.5% positive predictive value and 82.9% sensitivity in identifying the correct radiological features [28].

Our study has two limitations. First, this was a single-center design that employed only two raters, which may affect generalizability. Future multicenter studies with a larger pool of radiologists are recommended to enhance the external validity of retrospective case–control studies, which may be subject to a selection bias. Second, in our study population, the ratio of HCC to ICC cases differed from the actual clinical distribution.

Our study highlights the importance of multiphase CT imaging in differentiating HCC and ICC based on key radiological features and interobserver agreement among expert radiologists. The results indicate that arterial hyperenhancement, portal vein invasion, lymph node metastasis, and cirrhosis appearance are crucial imaging biomarkers that can aid in distinguishing these two malignancies. These findings provide a foundation for the development of AI models aimed at improving automated classification and reducing diagnostic variability.

Although this study does not involve direct AI model development, the results are intended to contribute to future AI training datasets by identifying reproducible imaging features with validated interobserver agreement, which could assist AI developers in refining liver cancer classification algorithms.

Given the promising role of AI in liver tumor differentiation, future studies should focus on integrating deep learning-based models trained on large, multi-center datasets to enhance generalizability. Additionally, further external validation of AI models is essential to ensure robust performance across diverse patient populations. Prospective studies should also be conducted to evaluate the real-world impact of AI-assisted diagnosis in clinical practice, including its potential role in LI-RADS classification, prognosis prediction, and treatment response assessment. Ultimately, incorporating AI into liver cancer diagnostics could enhance radiologist workflow efficiency, improve early detection, and contribute to personalized treatment strategies.

## 5. Conclusions

HCC and ICC share overlapping CT hallmark features, making differentiation challenging. This study highlights the role of ancillary imaging features, such as cirrhosis and lymph node metastasis, in improving diagnostic accuracy. By incorporating these features into AI training algorithms, AI-based models have the potential to enhance precision in distinguishing HCC from ICC.

As AI continues to evolve in radiology, future research should prioritize the development of large, multi-center datasets to ensure model generalizability and robustness.

These advancements could lead to more consistent diagnoses, improved early detection, and personalized treatment strategies for patients with liver malignancies.

## Figures and Tables

**Figure 1 diagnostics-15-00821-f001:**
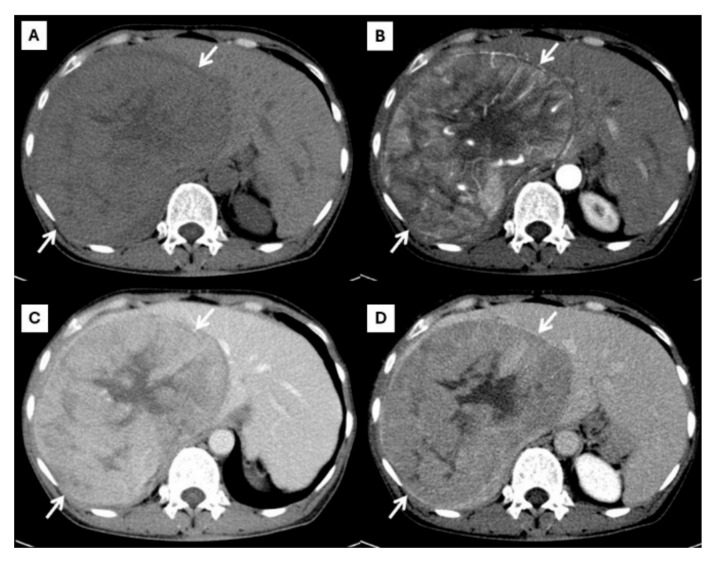
CT findings of typical HCC. Pre-contrast (**A**), arterial (**B**), portal venous (**C**), and delayed (**D**) post-contrast sequences demonstrate a mass with non-rim arterial hyperenhancement ((**B**), arrow), washout appearance in later phase ((**C**), arrow), and the presence of an enhancing capsule ((**D**), arrow).

**Figure 2 diagnostics-15-00821-f002:**
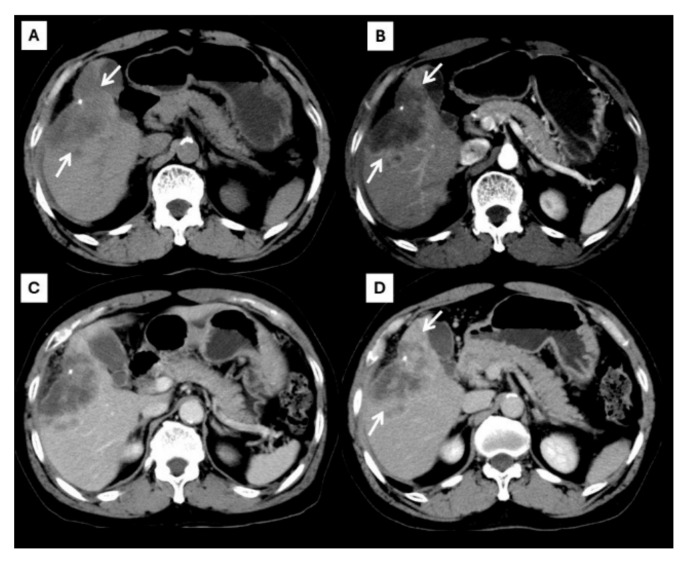
CT findings of typical CCA. Pre-contrast (**A**), arterial (**B**), portal venous (**C**), and delayed (**D**) post-contrast sequences demonstrate a mass with rim-like enhancement at the tumor periphery during the arterial phase ((**B**), arrow) and progressive centripetal enhancement in the later phase ((**C**,**D**), arrow).

**Figure 3 diagnostics-15-00821-f003:**
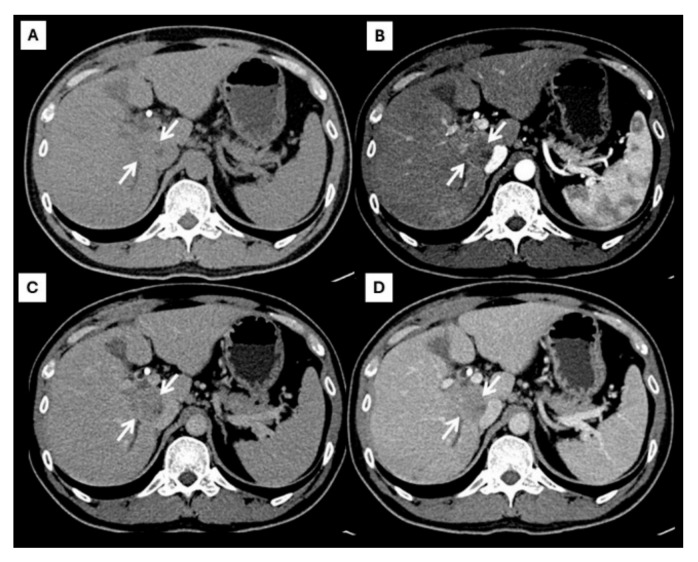
CT findings of atypical HCC. Pre-contrast (**A**), arterial (**B**), portal venous (**C**), and delayed (**D**) post-contrast sequences demonstrate a mass with rim-like enhancement at the tumor periphery during the arterial phase ((**B**), arrow) and progressive centripetal enhancement in the later phase ((**C**,**D**), arrow).

**Figure 4 diagnostics-15-00821-f004:**
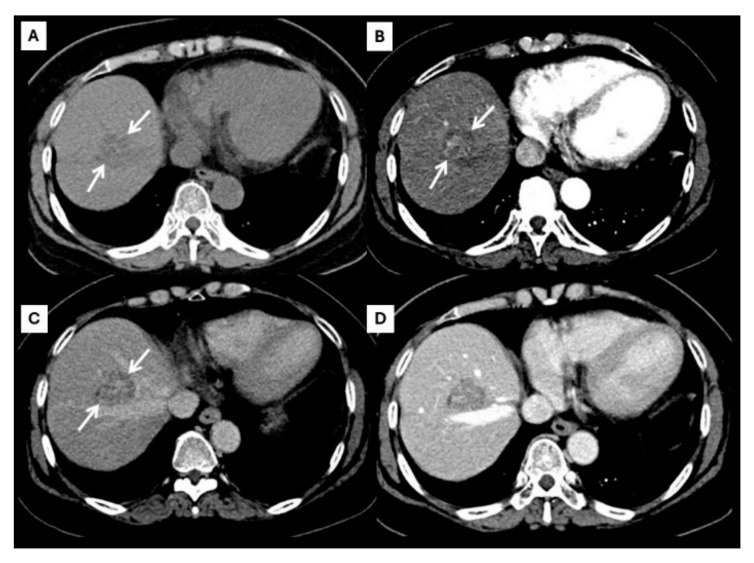
CT findings of atypical ICC. Pre-contrast (**A**), arterial (**B**), portal venous (**C**), and delayed (**D**) post-contrast sequences demonstrate a mass with non-rim arterial hyperenhancement ((**B**), arrow) and progressive centripetal enhancement in the later phase ((**C**,**D**), arrow).

**Figure 5 diagnostics-15-00821-f005:**
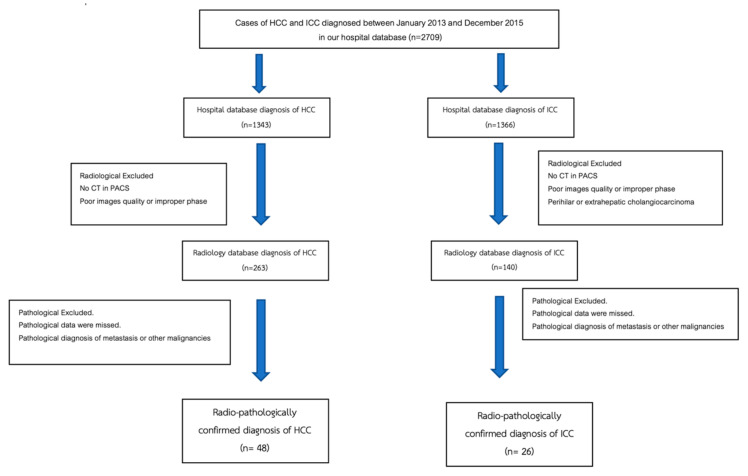
Flowchart of hepatocellular carcinoma and intrahepatic cholangiocarcinoma patient selection process.

**Table 1 diagnostics-15-00821-t001:** The characterization and difference of typical hepatocellular carcinoma and intrahepatic cholangiocarcinima detected using multiphase contrast-enhanced tomography.

	Hepatocellular Carcinoma	Intrahepatic Cholangiocarcinima
Later arterial phase	Arterial phase hyperenhancement	Rim-like enhancement
Portavenous phase	Non-peripheral washout	Progressive peripheral enhancement
Delayed phase	Capsule enhancement	Progressive peripheral enhancement

**Table 2 diagnostics-15-00821-t002:** Frequency of computed tomography (CT) features of histopathologically proven hepatocellular carcinoma (HCC) and intrahepatic cholangiocarcinoma (ICC) detected by expert radiologists.

	HCC (*n* = 48)	ICC (*n* = 26)
CT Features	Frequency	Frequency
Arterial enhancement		
Reader 1	95.8%	15.4%
Reader 2	95.8%	26.9%
Cirrhosis appearance		
Reader 1	70.8%	42.3%
Reader 2	72.9%	61.5%
Lymph node metastasis		
Reader 1	14.6%	50.0%
Reader 2	25.0%	57.7%
Portal vein invasion		
Reader 1	14.6%	11.5%
Reader 2	14.6%	19.2%

**Table 3 diagnostics-15-00821-t003:** Interobserver agreement among expert radiologists regarding computed tomography (CT) features of histopathologically proven hepatocellular carcinoma (HCC) and intrahepatic cholangiocarcinoma (ICC).

	HCC		ICC	
CT Features	Kappa	Agreement	Kappa	Agreement
Arterial enhancement	0.47	Moderate	0.66	Substantial
Cirrhosis appearance	0.43	Moderate	0.48	Moderate
Lymph node metastasis	0.29	Fair	0.84	Almost perfect
Portal vein invasion	0.66	Substantial	0.70	Substantial
Final diagnosis	0.65	Substantial	0.52	Moderate

**Table 4 diagnostics-15-00821-t004:** Diagnostic performance of two expert radiologists in the diagnosis of hepatocellular carcinoma (HCC) and intrahepatic cholangiocarcinoma (ICC).

	HCC				ICC			
Diagnosis	Sensitivity	Specificity	Positive Likelihood Ratio	Negative Likelihood Ratio	Sensitivity	Specificity	Positive Likelihood Ratio	Negative Likelihood Ratio
Reader 1	0.979	0.885	8.513043	0.024	0.885	0.979	42.142	0.117
Reader 2	0.958	0.731	3.561338	0.057	0.731	0.958	17.404	0.280

## Data Availability

The original contributions presented in this study are included in the article. Further inquiries can be directed to the corresponding author.

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
