# Peer review of "Multiphase Computed Tomography Scan Findings for Artificial Intelligence Training in the Differentiation of Hepatocellular Carcinoma and Intrahepatic Cholangiocarcinoma Based on Interobserver Agreement of Expert Abdominal Radiologists"

_diagnostics, 2025, doi:10.3390/diagnostics15070821_

Round 1
Reviewer 1 Report
Comments and Suggestions for Authors
This study analyzed CT images of HCC and ICC using a workstation, and correlated/assessed the agreement between radiologists, and then the correlation between CT imaging software and histological diagnosis. The text is well written, however the AI part of the analysis is not well explained. I understand that the fujifilm platform is used, however, there is no comparison between TC human only vs. AI-assisted. This should be explained well in abstract and rest of text as it may mislead the authors.
Additional comments:
(1) Regarding the differences between HCC and ICC. Could you please make a table and/or figure showing the characteristics and differences in the diagnosis. Please refer to https://radiologyassistant.nl/abdomen/liver/li-rads#li-rads-categories-li-rads-5-definitely-hcc as reference.
(2) Lines 103-112. Please revise the definitions of artificial intelligence, machine learning, deep learning, and ANNs; including supervised and unsupervised analysis. In the current text, it is not clear. Please refer to available literature. For example:
IBM documentation.
https://www.mdpi.com/journal/information/sections/Artificial_Intelligence
BioMedInformatics 2024, 4(2), 1480-1505; https://doi.org/10.3390/biomedinformatics4020081
(3) Line 122. Please add the WMA Helsinki statement for medical research.
https://www.wma.net/policies-post/wma-declaration-of-helsinki/
(4) Line 126. Regarding "confirmed diagnoses of HCC and ICC". Please explain how the diagnosis was confirmed. Was the diagnosis histopathological (gold standard) or based on other parameters (radiological)?
(5) Line 131. Could you please describe the histological criteria that differentitate HCC from ICC?
(6) In the current format, the letters of Figure 5 cannot be read properly. Please correct.
(7) Are the images of the two CT scanners 100% equal? Please be aware that a difference will create a bias in the image-base AI analysis.
(8) How many HCC and ICC were analyzed in each CT scanners? Were the samples equally distributed in the 2 equipment?
(9) Line 153. Please add the names of the 2 radiologists.
(10) Line 152 Please add the reference and website link for pacs and synapse workstation.
(11) Line 166. Please add the company and location of IBM SPSS software.
(12) Line 173. Regarding the Cohen's Kappa. Please confirm if "slight agreement" is 0.10 to 0.20.
(13) In results section, and line 279 "Artificial intelligence (AI) is a new tool in the radiology filed". Sorry to mention, but I saw no results of AI analysis in this paper. The correlation/agreement between radiologist is shown, but how the AI was conducted, results, performance paramenters are not shown (or I missed them). Of note, synapse radiology of fujifilm implements several AI imaging algorithms, but these are not described in this paper.
https://healthcaresolutions-us.fujifilm.com/products/enterprise-imaging/synapse-ai-orchestrator/
(14) How was the correlation/agreement between radiologists and AI-based analysis?
Author Response
(1) Regarding the differences between HCC and ICC. Could you please make a table and/or figure showing the characteristics and differences in the diagnosis. Please refer to https://radiologyassistant.nl/abdomen/liver/li-rads#li-rads-categories-li-rads-5-definitely-hcc as reference.
Response 1: Thank you for pointing this out. I agree with this comment. Therefore, I have made a table showing the characteristics and differences in the diagnosis of HCC and CCA, referring to https://radiologyassistant.nl/abdomen/liver/li-rads#li-rads-categories-li-rads-5-definitely-hcc . This change can be found on page number 2 and 3
(2) Lines 103-112. Please revise the definitions of artificial intelligence, machine learning, deep learning, and ANNs; including supervised and unsupervised analysis. In the current text, it is not clear. Please refer to available literature. For example:
IBM documentation.
https://www.mdpi.com/journal/information/sections/Artificial_Intelligence
BioMedInformatics 2024, 4(2), 1480-1505; https://doi.org/10.3390/biomedinformatics4020081
Response 2: Thank you for pointing this out. I agree with this comment. Therefore, I have added the details as the reference document . This change can be found on page number 3.
(3) Line 122. Please add the WMA Helsinki statement for medical research.
https://www.wma.net/policies-post/wma-declaration-of-helsinki/
Response 3: Thank you for pointing this out. I agree with this comment. Therefore, I have added add the WMA Helsinki statement for medical research. This change can be found on page number 5, line 142
(4) Line 126. Regarding "confirmed diagnoses of HCC and ICC". Please explain how the diagnosis was confirmed. Was the diagnosis histopathological (gold standard) or based on other parameters (radiological)?
Response 4: Thank you for pointing this out. The diagnosis of ICCs was confirmed by histopathology (biopsy or resection). The diagnosis of HCCs was confirmed by radiological findings (LR-5) and histopathology. This change can be found on page number 5, line 148-150
(5) Line 131. Could you please describe the histological criteria that differentitate HCC from ICC?
Response 5: The histological differentiation between hepatocellular carcinoma (HCC) and intrahepatic cholangiocarcinoma (ICC) is based on distinct pathological features:
- HCC is characterized by trabecular, pseudoglandular, or solid growth patterns, with tumor cells resembling hepatocytes. Additionally, HCC frequently shows sinusoidal capillarization and absence of desmoplastic stroma.
- ICC, on the other hand, is distinguished by glandular or ductal structures with abundant desmoplastic stroma. Tumor cells tend to be cuboidal or columnar with mucin production.
(6) In the current format, the letters of Figure 5 cannot be read properly. Please correct.
Response 6: Thank you for pointing this out. The new Figure 5 has replaced. This change can be found on page number 6.
(7) Are the images of the two CT scanners 100% equal? Please be aware that a difference will create a bias in the image-base AI analysis.
Response 7: Thank you for bringing this to my attention. I appreciate your insightful comment. The two CT scanners used in this study were different models but were manufactured by the same company. While they may not be completely identical, the differences are likely minimal and should not significantly impact the results.
(8) How many HCC and ICC were analyzed in each CT scanners? Were the samples equally distributed in the 2 equipment?
Response 8: I appreciate your comment and understand the importance of this information. Unfortunately, I do not have specific data on this aspect. However, as I mentioned in Response 7, the differences between the two scanners are minimal. Please let me know if any further clarification is needed.
(9) Line 153. Please add the names of the 2 radiologists.
Response 9: Thank you for pointing this out. I agree with this comment. Therefore, I have added the names of the 2 radiologists. This change can be found on page number 6, line 178-179.
(10) Line 152 Please add the reference and website link for pacs and synapse workstation.
Response 10: Thank you for pointing this out. I have added the reference and weblink for Synapse PACS. This change can be found on page number 6, line 176-177.
(11) Line 166. Please add the company and location of IBM SPSS software.
Response 11: Thank you for pointing this out. I have added company and location of IBM SPSS software. This change can be found on page number 7, line 193-194.
(12) Line 173. Regarding the Cohen's Kappa. Please confirm if "slight agreement" is 0.10 to 0.20.
Response 12: Thank you for pointing this out. I confirm that slight agreement is defined as a range from 0 to 0.20, so it includes values from 0.01 to 0.20.
(13) In results section, and line 279 "Artificial intelligence (AI) is a new tool in the radiology filed". Sorry to mention, but I saw no results of AI analysis in this paper. The correlation/agreement between radiologist is shown, but how the AI was conducted, results, performance paramenters are not shown (or I missed them). Of note, synapse radiology of fujifilm implements several AI imaging algorithms, but these are not described in this paper.
https://healthcaresolutions-us.fujifilm.com/products/enterprise-imaging/synapse-ai-orchestrator/
Response 13:
Thank you for your comment. The primary objective of this research is to establish a structured dataset based on radiological features that could later be utilized in AI model development. Therefore, while the correlation and agreement between radiologists are presented in this study, AI analysis, results, and performance parameters have not yet been included.
We acknowledge AI-assisted tools like Fujifilm Synapse Radiology, but as our study does not yet involve AI implementation, such algorithms were not included.
(14) How was the correlation/agreement between radiologists and AI-based analysis?
Response 14: Thank you for your question. At this stage, we do not yet have AI-based analysis. This study focuses on identifying radiological features for future AI training, and the correlation/agreement presented is based solely on radiologist interpretations.
Reviewer 2 Report
Comments and Suggestions for Authors
Review Report for MDPI Diagnostics
(Multiphase CT scans findings for AI training in the differentiation of hepatocellular carcinoma and intrahepatic cholangiocarcinoma based on interobserver agreement of expert abdominal radiologist)
1. Within the scope of the study, approximately three years of data on patients diagnosed with intrahepatic cholangiocarcinoma and hepatocyte carcinoma liver cancer were used to analyze computerized tomography scans in relation to artificial intelligence.
2. In the introduction, information on hepatocyte carcinoma and intrahepatic cholangiocarcinoma and computerized tomography findings, the importance of the subject and the place of artificial intelligence in the literature on this subject were mentioned at a basic level. It is definitely recommended to detail the literature review in this section.
3. In the study, liver cancer computerized tomography images, which received ethics committee approval and were collected specifically for the study, were used as a dataset. The amount and distribution of the dataset seem to be sufficient for this study.
4. When the computerized tomography features associated with both types of liver cancer detected by expert radiologists are examined in detail, it is understood that they are at an acceptable level in terms of both feature type and kappa values.
5. When the sensitivity and spec values obtained by the expert radiologist for both intrahepatic cholangiocarcinoma liver cancer and hepatocyte carcinoma liver cancer are examined in terms of literature, it is seen that they are at an appropriate level.
6. The conclusion section is very limited in terms of the study and it is recommended to be detailed more and to be addressed more deeply in terms of future studies.
In conclusion, the study has the potential to contribute to the literature in terms of liver cancer studies. However, attention should be paid to the sections mentioned above.
Author Response
- In the introduction, information on hepatocyte carcinoma and intrahepatic cholangiocarcinoma and computerized tomography findings, the importance of the subject and the place of artificial intelligence in the literature on this subject were mentioned at a basic level. It is definitely recommended to detail the literature review in this section.
Response 1. Thank you for your recommendation. I have revised the introduction part as follows
AI models have been trained on CECT datasets to extract key imaging biomarkers that help distinguish HCC from ICC and other liver lesions. Several studies have demonstrated the effectiveness of AI in liver tumor classification. Yasaka et al. developed CNN model capable of differentiating HCC, ICC, hemangiomas, and metastases using multiphase CT, achieving a 92% area under the receiver operating characteristic curve (AUC)(18). Similarly, Ponnoprat et al. utilized deep learning techniques on multiphase CT scans, reporting an 88% classification accuracy in distinguishing HCC from ICC(18). Wang et al. further advanced AI applications by proposing an interpretable radi-omics-based system that identifies radiological features in liver tumors, demonstrating an 82.9% sensitivity in detecting HCC(19).
This change can be found on page number 5, line 126-135.
- The conclusion section is very limited in terms of the study and it is recommended to be detailed more and to be addressed more deeply in terms of future studies.
Response 2. I appreciate the reviewer’s suggestion and have expanded the Conclusion section to provide a more comprehensive summary of our findings and discuss potential directions for future research as follows.
Our study highlights the importance of multiphase CT imaging in differentiating HCC and ICC based on key radiological features and interobserver agreement among expert radiologists. The results indicate that arterial hyperenhancement, portal vein invasion, lymph node metastasis, and cirrhosis appearance are crucial imaging biomarkers that can aid in distinguishing these two malignancies. These findings provide a foundation for the development of AI models aimed at improving automated classification and reducing diagnostic variability.
Given the promising role of AI in liver tumor differentiation, future studies should focus on integrating deep learning-based models trained on large, multi-center datasets to enhance generalizability. Additionally, further external validation of AI models is essential to ensure robust performance across diverse patient populations. Prospective studies should also be conducted to evaluate the real-world impact of AI-assisted diagnosis in clinical practice, including its potential role in LI-RADS classification, prognosis prediction, and treatment response assessment. Ultimately, incorporating AI into liver cancer diagnostics could enhance radiologist workflow efficiency, improve early detection, and contribute to personalized treatment strategies.
This change can be found on page number 9-10, line 299-315.
Reviewer 3 Report
Comments and Suggestions for Authors
Manuscript title
“Multiphase CT scans findings for AI training in the differentiation of hepatocellular carcinoma and intrahepatic cholangiocarcinoma based on interobserver agreement of expert abdominal radiologist.”
- The main research question addressed by the presented manuscript includes investigation of potential additional computed tomography features (arterial hyperenhancement, cirrhosis, lymph node metastasis, portal vein invasion) and their interobserver agreement to differentiate two primary liver malignancies (hepatocellular and cholangiocellular carcinoma), aiming to improve artificial intelligence (AI) algorithms.
- This work may be considered relevant to AI quality assurance in radiology, as it identifies possible additional imaging features to enhance AI model accuracy in distinguishing HCC and ICC. However, the authors do not develop an original model to show that inclusion of additional features would indeed increase the diagnostic performance.
- The authors cite existing LI-RADS criteria and prior AI/ML studies, proposing ancillary CT features (cirrhosis and lymph node metastasis) for future AI training.
- Specific improvements to study methodology include:
- Adherence to the GRASS checklist (https://www.sciencedirect.com/science/article/abs/pii/S0020748911000368) would be most welcome to increase study replicability and scientific rigor
- Providing additional information on evaluation criteria (binary? quantitative? 4-point scale? reader experience? number of sessions?)
- Including information on image reconstruction protocol (kernel? slice thickness?)
- Elaborating on inter-reader diagnostic performance (differences in accuracy statistically significant?)
- Revising “Discussion” section, as currently it only partially aligns with the journal guidelines and expanding on possible implications of including features with moderate agreement between readers (i.e., cirrhosis)
- The conclusions may require revision after addressing the GRASS checklist.
- The references are appropriate.
- Figures are appropriate. Table 3 would benefit from using standard terminology (i.e., specificity instead of “spec”).
Consider additional proofreading using Grammarly or similar tool to correct typos and improve text readability, including in Figures.
Author Response
2. This work may be considered relevant to AI quality assurance in radiology, as it identifies possible additional imaging features to enhance AI model accuracy in distinguishing HCC and ICC. However, the authors do not develop an original model to show that inclusion of additional features would indeed increase the diagnostic performance.
Response 1: Thank you for your comment. This study focuses on identifying radiological features for AI training, rather than presenting AI training model.
- Providing additional information on evaluation criteria (binary? quantitative? 4-point scale? reader experience? number of sessions?)
Response 2: Thank you for your recommendation. I revised the manuscripts as per your suggestion. This change can be found on page number 6.
- Including information on image reconstruction protocol (kernel? slice thickness?)
Response 3: Thank you for your recommendation. I had added “The images were reconstructed in 2-mm slice thickness in axial, coronal, and sagittal views.” This change can be found on page number 6, line 172.
- Elaborating on inter-reader diagnostic performance (differences in accuracy statistically significant?)
Response 4: The diagnostic performance of the two radiologists was satisfactory with a corrected diagnosis of 87.8% and 94.6%, respectively. There is no significant difference.
- Revising the “Discussion” section, as currently it only partially aligns with the journal guidelines and expanding on possible implications of including features with moderate agreement between readers (i.e., cirrhosis)
Response 5: Thank you for your recommendation. I will follow your suggestion. This change can be found on page number 9-10, line 299-315.
The conclusions may require revision after addressing the GRASS checklist.
Response 6. I appreciate the reviewer’s suggestion and have expanded the Conclusion section to provide a more comprehensive summary of our findings and discuss potential directions for future research as follows.
HCC and ICC share overlapping CT hallmark features, making differentiation challenging. This study highlights the role of ancillary imaging features, such as cirrhosis and lymph node metastasis, in improving diagnostic accuracy. By incorporating these features into AI training algorithms, AI-based models have the potential to enhance precision in distinguishing HCC from ICC.
As AI continues to evolve in radiology, future research should prioritize the development of large, multi-center datasets to ensure model generalizability and robustness.
These advancements could lead to more consistent diagnoses, improved early detection, and personalized treatment strategies for patients with liver malignancies.
This change can be found on page number 10, line 318-326.
Figures are appropriate. Table 3 would benefit from using standard terminology (i.e., specificity instead of “spec”).
Response 7: Thank you for your recommendation. I have edited the table as per your suggestion. This change can be found on page number 8.
Round 2
Reviewer 1 Report
Comments and Suggestions for Authors
Thank you very much for the answers. Now it is clearer. If AI methodology was not applied, my recommendation is to please remove from the title "for AI training" as this creates confusion to the readers. In the conclusion, I would also remove "adopted in future AI training algorithms" because no AI implementation has been performed. Note: The Fuji PACS and using Synapse workstation may be using some kind of AI-based implementation, but this is not explained/described deeply in the paper.
Author Response
Thank you for your thoughtful comment. While we acknowledge that AI methodology has not been directly implemented in this study, we would like to respectfully retain both the title and the statement in the conclusion. The primary aim of our study was to identify and validate radiological features that could serve as a foundational dataset for future AI training. We believe that the findings from this research, particularly the interobserver agreement on key imaging features, will be valuable for AI developers working on training algorithms for differentiating HCC from ICC.
Although AI was not applied within this specific study, the clinical relevance and reproducibility of the reported features are intended to support future AI model development. We hope this clarifies our rationale, and we are open to adding a clarifying sentence in the Discussion section to address this point directly.
Although this study does not involve direct AI model development, the results are intended to contribute to future AI training datasets by identifying reproducible imaging features with validated interobserver agreement, which could assist AI developers in refining liver cancer classification algorithms.
You can see this change in page 10 with blue color.
Reviewer 3 Report
Comments and Suggestions for Authors
The authors provided revisions, further improving the manuscript.
Response 1: not accepted, as if authors do not plan to develop an artificial intelligence algorithm in a future study, the current result's relevance may not be readily apparent to the reader. Consider clearly stating future study directions.
Response 2: not accepted. What were the formal criteria for arterial hyperenhancement, portal vein invasion, cirrhosis, and lymph node metastasis? Were they defined?
Response 3: accepted.
Response 4: not accepted. What statistical test was used to compare accuracies on paired data? Report it as well as the p-value.
Response 5: accepted.
Response 6: not accepted, as the revised manuscript does not adhere to the GRASS checklist for reliability studies, limiting its replicability and potential impact.
Response 7: accepted.
Additionally, the English proofreading had not been performed.
Comments on the Quality of English LanguageConsider additional proofreading using Grammarly or similar tool to correct typos and improve text readability, including in Figures.
Author Response
Response 1:
Thank you for your thoughtful comment. While we acknowledge that AI methodology has not been directly implemented in this study, we would like to respectfully retain both the title and the statement in the conclusion. The primary aim of our study was to identify and validate radiological features that could serve as a foundational dataset for future AI training. We believe that the findings from this research, particularly the interobserver agreement on key imaging features, will be valuable for AI developers working on training algorithms for differentiating HCC from ICC.
Although AI was not applied within this specific study, the clinical relevance and reproducibility of the reported features are intended to support future AI model development. We hope this clarifies our rationale, and we are open to adding a clarifying sentence in the Discussion section to address this point directly.
Although this study does not involve direct AI model development, the results are intended to contribute to future AI training datasets by identifying reproducible imaging features with validated interobserver agreement, which could assist AI developers in refining liver cancer classification algorithms.
You can see this change in page 10 with blue color.
Response 2: Thank you for your recommendation. I revised the manuscripts as per your suggestion. This change can be found on page number 6 and 7.
Response 4: I added the p value in manscripts. (MaNemar ‘s chi2, p value 0.108). This change can be found on page number 8.
Response 6. Thank you for your comment and for highlighting the importance of the GRASS checklist for reliability studies. We acknowledge the value of adhering to reporting guidelines to enhance transparency and replicability. However, due to certain limitations in our study, we were unable to fully meet all GRASS checklist items.
Nevertheless, we have made efforts to address key elements of the checklist, including clearly defining imaging criteria, describing the rater characteristics, and reporting the methods of interobserver agreement. This change can be found on page number 6, 7 and 10 in blue color.
We appreciate your understanding.
Additionally, the manuscript has been proofread by our English language center. However, we are happy to conduct another round of proofreading prior to publication, if necessary.
Round 3
Reviewer 3 Report
Comments and Suggestions for Authors
The authors have provided sufficient responses to reviewer's suggestions.
Comments on the Quality of English LanguageConsider proofreading the abstract and main text in order to eliminate the remaining errors (singular/plural, incorrect tense, suboptimal prepositions) and improve the readability.
For abstract at least, consider the proposed revisions (underlined):
- The purpose of this study was to find out potential CT features for the differentiation of hepatocellular carcinoma and intrahepatic cholangiocarcinoma.
- The arterial hyperenhancement, portal vein thrombosis, lymph node enlargement, and cirrhosis appearance were evaluated. We then calculated sensitivity, specificity, and likelihood ratio for diagnosis of HCC and ICC.
- The diagnostic performance of the two radiologists was satisfactory, with the correct diagnosis of 87.8% and 94.6%.
- Two radiologists had high sensitivity in diagnosing HCCs (95.8% to 97.9%) and specificity in diagnosing ICCs (95.8% to 97.9%).
Author Response
Thank you very much for your insightful and constructive comments. I greatly appreciate your suggestions and will ensure they are thoroughly addressed. I have changed it to green.